# Copilot Evaluation Harness: Building User Trust in LLMs and LM Agents for IDE Environments

## Abstract

The addition of Large Language Models (LLMs) into Integrated Code Development Environments (IDEs) has become a focal point in modern software development. LLMs offer the potential to significantly augment developer productivity by serving as intelligent, chat-driven programming assistants, especially with the increase in LLM-driven coding agents. With these tools comes the need for safeguards and metrics for quality assurance for consumers. In this paper, we introduce the Copilot Evaluation Harness: a set of data and tools for evaluating LLM-guided coding, covering various programming scenarios and languages. We propose a more robust system for measuring and understanding model behavior when leveraged as chat coding assistants or coding agents than previous state of the art evaluation metrics. We design and compute both static and execution-based success metrics on a wide range of developer tasks, including documentation generation from code (doc), test case generation (test), and bug-fixing (fix). In the chat scenario, we see that GPT4o has much lower prompt sensitivity than the other models. In the agentic scenario, we find that reasoning models are more inclined to generate one-shot solutions, even when given multiple turns and access to tool calling. We show how results from our metrics can be used to increase the interpretability and explainability of LLMs in the real-world IDE-chat scenario.

## 1 Introduction

The continuous evolution of software development practices has led to a growing interest in the integration of cutting-edge technology to enhance developer productivity Chen et al. (2021). One such technology that has garnered considerable attention is the utilization of Large Language Models (LLMs) within Integrated Development Environments (IDEs) Nam et al. (2023); Chen et al. (2023), such as VSCode Copilot Chat or Cursor AI. Large language models, exemplified by models like OpenAI's GPT-4o and Anthropic's Sonnet 3.5, as well as open-source models such as DeepSeek V3, offer the promise of acting as intelligent programming assistants and agents OpenAI (2023) OpenAI (2024) Anthropic (2024) DeepSeek-AI & et al (2024). In this paper, we introduce the Copilot Evaluation Harness for comprehensive exploration of the behaviors of LLMs as both coding assistants and coding agents, with a particular focus on verifying their trustworthiness across diverse programming scenarios and languages. Figure 1 shows a high-level overview of the phases of our evaluation.

Our harness evaluates four major software development scenarios: bug fixing, documentation generation, method generation, and test generation. These scenarios encompass a spectrum of developer tasks, each addressing specific challenges and opportunities.

Previous evaluation harnesses leave gaps we seek to cover with the Copilot Evaluation Harness. In the HumanEval dataset Chen et al. (2021), for example, models are evaluated on their ability to generate functions from docstrings for straightforward, single-file algorithmic questions. While benchmarks such as SWE-Bench Jimenez et al. (2024) work with real world code in complex scenarios, they focus on batch-oriented Pull Request workflows rather than interactive IDE scenarios where agents can leverage real-time diagnostic tools and context discovery features. With Copilot Evaluation Harness, we take these principles and expand upon them: like SWE-Bench, our evalua-

Figure 1: Overview of the Copilot Evaluation Harness architecture. The user interaction component is brought by the harness users. This diagram shows a sample user interaction, but the user interaction can be as simple as a single model call. The evaluation harness supports data collection and metric evaluation.

tion expects the model to interact with real-world repositories which involve dozens of methods and files to complete a given task. Unlike SWE-Bench, we do not evaluate a model or an agent's ability in a Pull Request workflow. Instead, we evaluate IDE-integrated coding assistants and agents, with full access to user's workspace and more chat-driven instructions. Therefore, we fill the current lack of robust evaluations for this real-world setting in IDE.

We apply our evaluation framework within the LLM-powered chat extension in Visual Studio Code (VS Code), an IDE used by 15 million programmers across the world. Our evaluation spans a spectrum of language models, ranging from proprietary models like OpenAI's GPT-4o and Anthropic's Sonnet 3.5 to openly available alternatives such as DeepSeek v3.

We find that the models perform comparably in the chat scenario, with test generation being the most difficult task. More advanced models such as o3-mini are much more sensitive to the wording and location of information in the prompt, whereas GPT4o often outright ignores many instructions. Import statement generation proves to be a challenging component of test generation, especially for Python cases.

In addition to the vanilla chat scenario, we evaluate an agentic flow with a basic agent setup. Here, we see that o3-mini leverages its reasoning ability to turn the agentic process into a one-shot process, attempting to complete the task in only one turn, even in cases where Sonnet 3.5 and GPT4o take five to ten turns. While Sonnet 3.5 and GPT4o call tools to get errors in the current file or search the given codebase, o3-mini will take the input file and query prompt and attempt a solution without calling any tools besides file editing.

Use of the Copilot Evaluation Harness enables a new level of understanding of model behavior that is beyond usual numeric values shared in public benchmarks. It allows engineers to update their integrations for increased quality of model responses, leading to greater user trust and satisfaction.

## 2    COPILOT EVALUATION HARNESS

Copilot Evaluation Harness is a set of four benchmark metrics created from public Github projects in six programming languages. The task for each metric is as follows:

- **Documentation Generation from Code** (doc): the model is given a file and the line range of a function. It must insert a docstring for the function with correct syntax, and with all the parameters and returned objects defined.
- **Bug-Fixing** (fix): given a static analyzer error and a line range, the model must output a patch on the line range to fix the error.
- **Code Generation from Natural Language** (generate): given a file that contains a function signature with its body missing, the model must fill in the function body correctly. Correctness is measured by running the repository's test suite and confirming that the tests that cover the function all pass.
- **Test Case Generation for Code** (test): the model must generate a test suite for a given function. The test suite is run in the repository environment and scored based on whether the generated tests pass.

## 2.1 Repository Collection

Our dataset is made up of methods from 300 public GitHub repositories across five languages: JavaScript, Typescript, Python, Java, and C#. We filter repositories that are smaller than 1 MB and larger than 100 MB. We also filter repositories for which it takes longer than 10 minutes to build and run tests. Language-specific heuristics for collected repositories can be found in the appendix section C.

As part of our evaluation harness, we have developed a build agent that utilizes various build and test strategies on any arbitrary repository. In addition, we have the capability to run static analysis tools on the repositories that we can build and test. This build agent is essential in collecting the test datasets and performing evaluations. The code for our build agent, evaluation scripts and data can be found at `https://anonymous.4open.science/r/copilot-evaluation-harness-38E5/README.md`.

## 2.2 Metric Constructions

After identifying suitable repositories for each language, we generate test cases for each evaluation metric based on the code within the repositories. Most evaluations require identifying methods that meet certain conditions, such as being covered by existing tests or containing a warning from a static analysis tool. The criteria for generating evaluation test cases varies from metric to metric, and is explained for each metric below. Additional details can be found in the Appendix section D.

**Documentation Generation from Code (doc).** We create test cases by identifying methods in the repository that are longer than three lines and are not a result of minification or obfuscation. We provide the method and ask the model to generate a docstring for the method. We consider a docstring generation to be successful if the location, format, and coverage of the generated text is correct.

**Bug Fixing (fix).** We create test cases based on static analysis tool warnings and errors flagged on a given repository. We run the static analysis tools within the context of a repository's built state (i.e. the virtual environment for Python, after running npm install for Javascript and Typescript). We consider a generated fix to be successful if it is syntactically correct and strictly reduces the numbers of static analysis warnings on execution. We must consider a strict decrease rather than the presence of the original warning or error, because it is possible for the coding assistant to fix the original issue while introducing a new issue, which a developer would not look upon as a complete fix.

**Code Generation from Natural Language (generate).** We select test cases by identifying methods in a given repository that are covered by a passing test in the repository's test suite. We formulate the task by removing the *body*, but not the signature of a method, then passing the altered file to the model. We consider a generated code snippet to be successful if the generated code is syntactically correct and all test cases covering the generated code pass.

**Test Generation from Code (test).** We create test cases by identifying methods within a given repository. We ask the model to provide a test suite for the given method. We consider the generation to be successful if it invokes the given method, and if the generated suite passes upon execution.

## 3 Experimentation and Analysis

We use the VSCode IDE to evaluate three metrics on myriad LLMs to gauge reliability and correctness of coding assistant-generated code in two scenarios: chat and agentic. Results for method generation will be included once the data for all languages has been finalized.

### 3.1 Chat Scenario

**Problem Formulation.** In the chat scenario, we provide the model the contents of the relevant file for the task, as well as a one-line query specifying what the models needs to do for the metric. We give the model one response to fulfill the needs of the given metric. Table 1 shows the results by language for DeepSeek V3, GPT4o, o3-mini, and Sonnet 3.5.

| Suite | Language | DeepSeek v3 | GPT-4o | o3 mini | Sonnet 3.5 |
|---|---|---|---|---|---|
| **DOC** | C# | 84% | 86% | 82% | 82% |
| | Java | 99% | 98% | 98% | 99% |
| | Javascript | 73% | 69% | 71% | 71% |
| | Python | 88% | 95% | 88% | 88% |
| | Typescript | 90% | 95% | 89% | 86% |
| **FIX** | C# | 78% | 76% | 79% | 78% |
| | Java | 45% | 43% | 58% | 59% |
| | Javascript | 63% | 81% | 74% | 94% |
| | Python | 92% | 94% | 97% | 94% |
| | Typescript | 95% | 90% | 93% | 95% |
| **TEST** | C# | 15% | 15% | 13% | 23% |
| | Java | 44% | 48% | 43% | 45% |
| | Javascript | 16% | 22% | 20% | 26% |
| | Python | 9% | 18% | 23% | 22% |
| | Typescript | 14% | 12% | 28% | 13% |

Table 1: Chat scenario comparison of DeepSeek V3, GPT4o, o3-mini, and Sonnet 3.5 models across various programming languages and tasks. Each number is reported on the same random sample of 100 test cases per language per metric

**Model Analysis.** All four evaluated models show promising results on our metrics, with test generation proving to be the most difficult. Further analysis into the largest discrepancies between models leads to a few key findings. First, we see that GPT4o's decreased sensitivity to prompt instructions also makes it more immune to over-fitting to code examples in the prompt. In the bug fixing scenario, the prompt includes a small patch of sample code at the end of the prompt to help guide the model's eye. This code block does not appear verbatim in the code, but is a similar structure to the code block the model must fix. Figure 9 in Appendix E shows the discrepancy between the two code blocks. DeepSeek V3 and o3-mini were very negatively impacted due to this diagnostic code: rather than applying a patch to the correct code block, they tried to apply the patch for the diagnostic code block, which was only included as a potentially relevant example. GPT4o, with its decreased prompt sensitivity, did not experience this issue. Sonnet 3.5 had the same issue as DeepSeek V3 and o3-mini, but less frequently, allowing it to still outperform the other models.

In the Python test generation scenario, a large source of errors comes from model's attempting to generate import statements. The prompt specifies the location to which the generated test suite will be written, and the models must write import statements accordingly. We see that o3-mini is much more likely to generate correct import statements than GPT4o, with Sonnet 3.5 and DeepSeek V3 in between the two.

## 3.2 Agentic Scenario

**Problem Formulation.** In the agentic scenario, we evaluate the metrics using a basic agent flow with the following tools: edit file, search file, get errors, read file, search codebase, and some additional terminal commands. We give the models multiple turns to solve each case. Unlike the chat scenario, we do not provide a code selection in the model input. We also do not include the details of the static analyzer error for bug fixing. Rather, the agent must discover and solve the error autonomously. Table 2 shows the performance of GPT4o, o3-mini and Sonnet 3.5 in the agentic flow.

**Model Analysis.** Figures 2, 3 and 4 display agentic model trajectories. The x-axes represent the step number. Each color in the stacked bar represents a different function called by the given model on the given step. These plots highlight an interesting phenomenon:

| Suite | Language | GPT-4o | o3-mini | Sonnet 3.5 |
|---|---|---|---|---|
| | C# | 67% | 79% | 78% |
| | Java | 52% | 56% | 69% |
| **DOC** | Javascript | 55% | 56% | 59% |
| | Python | 93% | 90% | 90% |
| | Typescript | 71% | 66% | 75% |
| | C# | 80% | 31% | 76% |
| | Java | 71% | 15% | 66% |
| **FIX** | Javascript | 91% | 82% | 82% |
| | Python | 98% | 95% | 100% |
| | Typescript | 96% | 97% | 94% |
| | C# | 17% | 21% | 17% |
| | Java | 60% | 62% | 53% |
| **TEST** | Javascript | 23% | 20% | 17% |
| | Python | 36% | 56% | 38% |
| | Typescript | 18% | 20% | 18% |

Table 2: Agentic scenario comparison of GPT4o, o3-mini, and Sonnet 3.5 models across various programming languages and tasks.

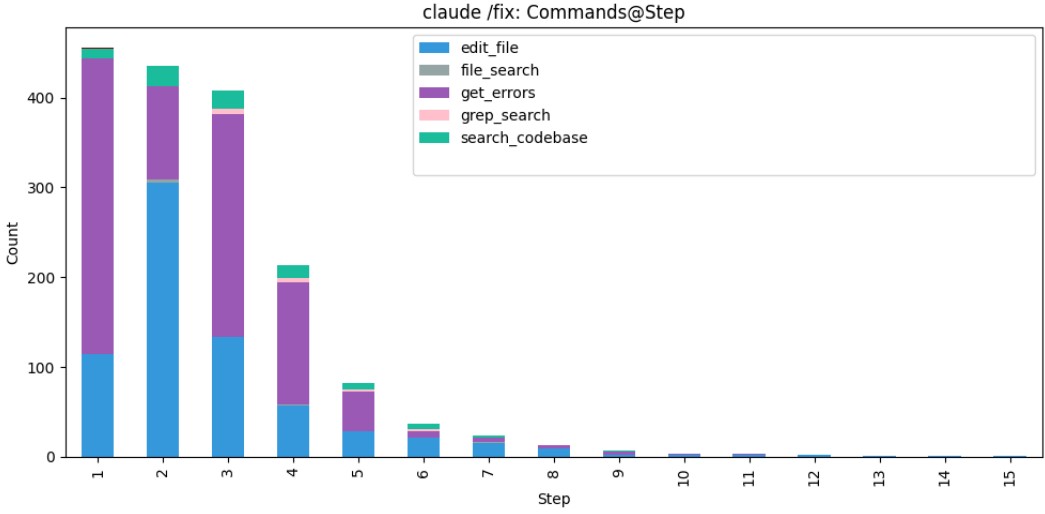

Figure 2: Sonnet 3.5 trajectory with basic agent for the bug fixing scenario.

**O3-mini skips the error identification step, leading to faster but less accurate solutions**. o3-mini does not call the *get_errors* function, while GPT4o and Sonnet 3.5 consistently call it in the first few steps. Although none of the models are presented with the specifics of the error, o3-mini assesses the given file to predict what the error might be, rather than explicitly calling the relevant tool. This leads to fewer overall turns, but often worse results: it thinks it has solved the problem without verifying what the problem is. This finding about the o3-mini model is not possible to ascertain with a benchmark like SWE-bench, because SWE-bench always gives the model the issue it needs to fix. In contrast, we expect the agentic flow to find the error with tool calling before fixing it.

**Reasoning-focused o3-mini favors immediate test generation versus an iterative approach**. o3-mini tends to treat the agentic, multi-turn process as a one-shot task for test generation as well.

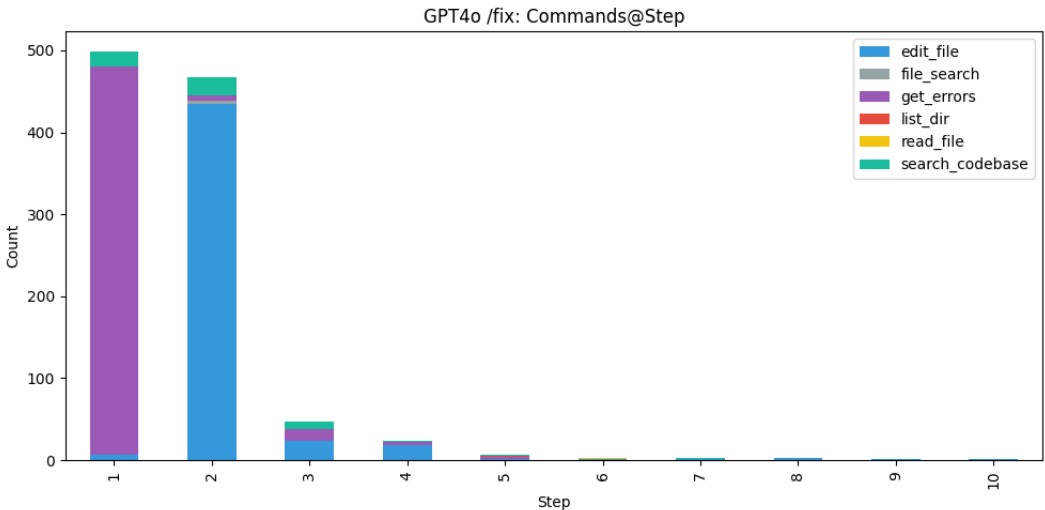

Figure 3: GPT4o trajectory with basic agent for the bug fixing scenario.

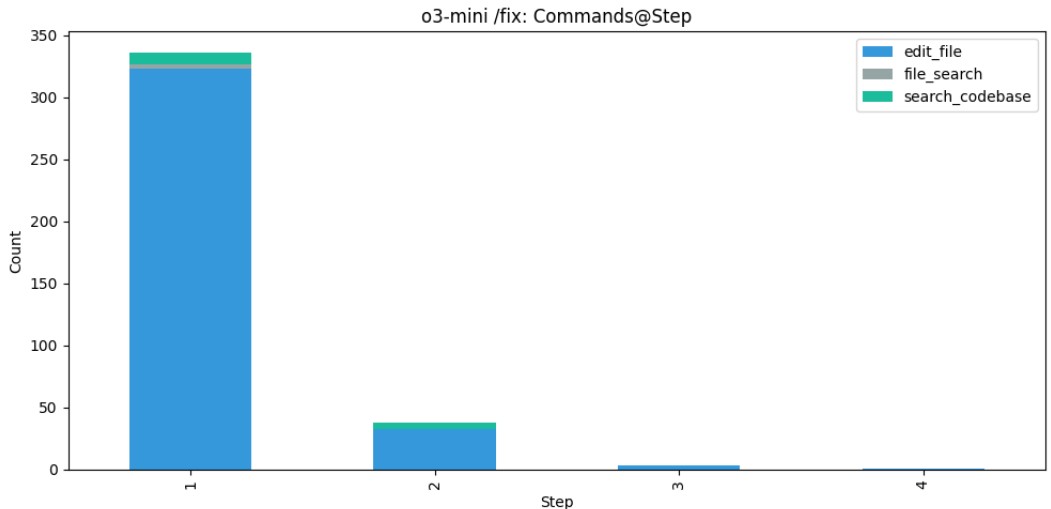

Figure 4: o3-mini trajectory with basic agent for the bug fixing scenario.

While GPT4o and Sonnet 3.5 generally take two to three turns to write the test, o3-mini almost always writes the tests immediately without any additional tool calling. Figure 5 highlights this difference.

**Sonnet 3.5 exercises more cautiousness by iterative error checking after editing.** In the documentation generation scenario, Figure 6 shows how Sonnet 3.5 always calls *get_errors* after adding the docstring to the file. It does this to confirm that it hasn't introduced errors in the process of adding the docstring. In comparison, GPT 4o and o3-mini respond quite similarly to each other: edit the file in one turn and mark the task as complete. Similarly in bug fixing, Sonnet 3.5 is more likely to call *get_errors* again in subsequent steps after editing the file.

## 4 RELATED WORK

We explain how our work builds upon and extend the related work on LLMs, Evaluating LLMs, and Evaluating LLMs for software engineering tasks.

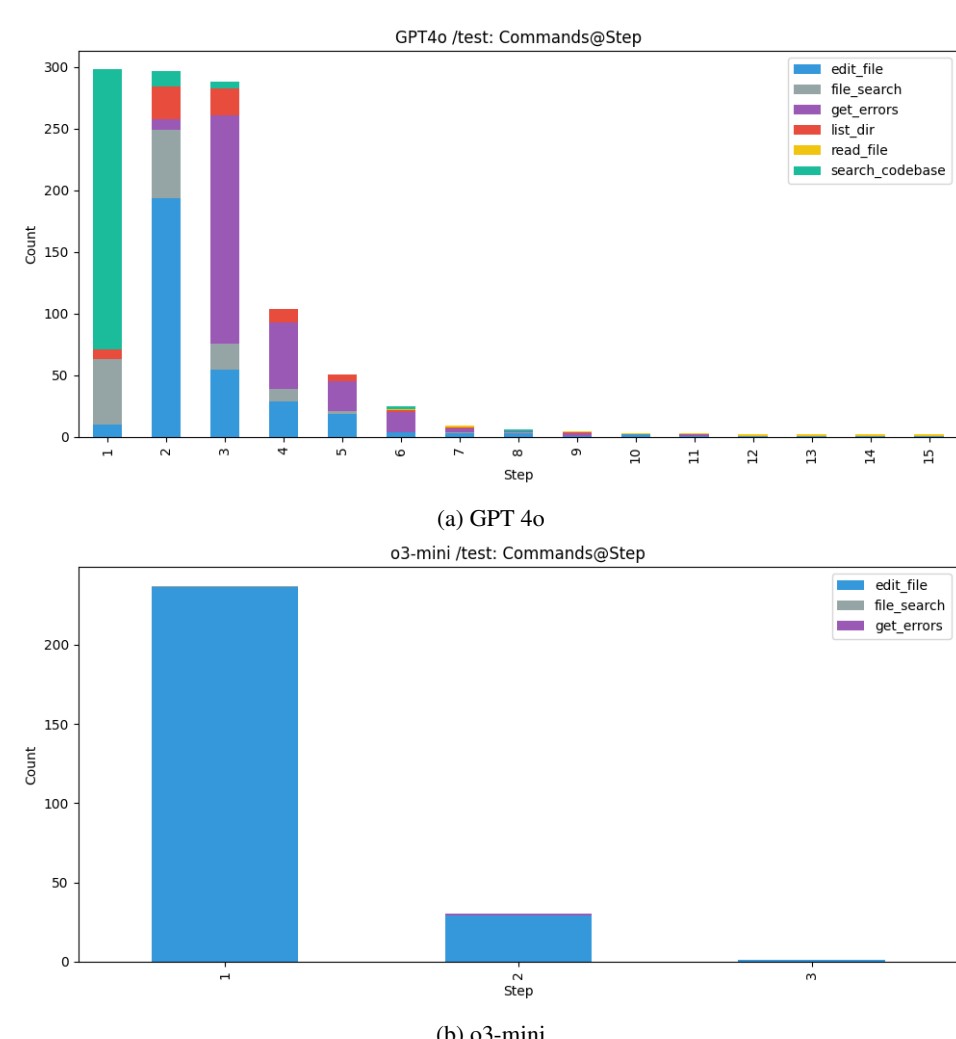

(a) GPT 4o

(b) o3-mini

Figure 5: GPT4o vs. o3-mini agentic trajectories for the test generation scenario.

**LLMs.** Building on the success of LLMs, researchers have started to explore the advantages of scaling up LLMs. For example, Gropher Rae et al. (2022) has 280 billion parameters, Megatron-turing NLG Smith et al. (2022) has 530 billion parameters and PaLM Chowdhery et al. (2022) has 540 billion parameters outperforming average humans on the BIGbench benchmark Srivastava et al. (2023). Similarly, researchers also explored fine-tuning LLMs for specific tasks and/or with human feedback Ouyang et al. (2022). In our comprehensive study, we examine the performance of four prominent LLMs: OpenAI's GPT-4o and o3-mini, Anthropic's Sonnet 3.5 and DeepSeek V3 on multiple software engineering scenarios.

**Evaluating LLMs.** Previous work has evaluated the effectiveness of LLMs including performance in natural language tasks, reasoning, robustness, safety, etc Chang et al. (2023). For tasks like sentiment analysis, Liang et al. (2023) and Qin et al. (2023) showed that LLMs perform much better than traditional ML models. Other works Laskar et al. (2023) have evaluated ChatGPT's performance on a range of tasks including answering questions, text summarization, code generation, reasoning, and addressing ethical issues. Unlike traditional machine learning models where k-fold cross validation was a common evaluation process, LLMs are often evaluated using static data sets. Common datasets for evaluating LLMs include: GLUE Wang et al. (2019b), SuperGLUE Wang et al. (2019a), BIGBench Srivastava et al. (2023), Massive Multitask Language Understanding (MMLU) Hendrycks et al. (2021), Ethics Benchmark Hendrycks et al. (2023), and others.

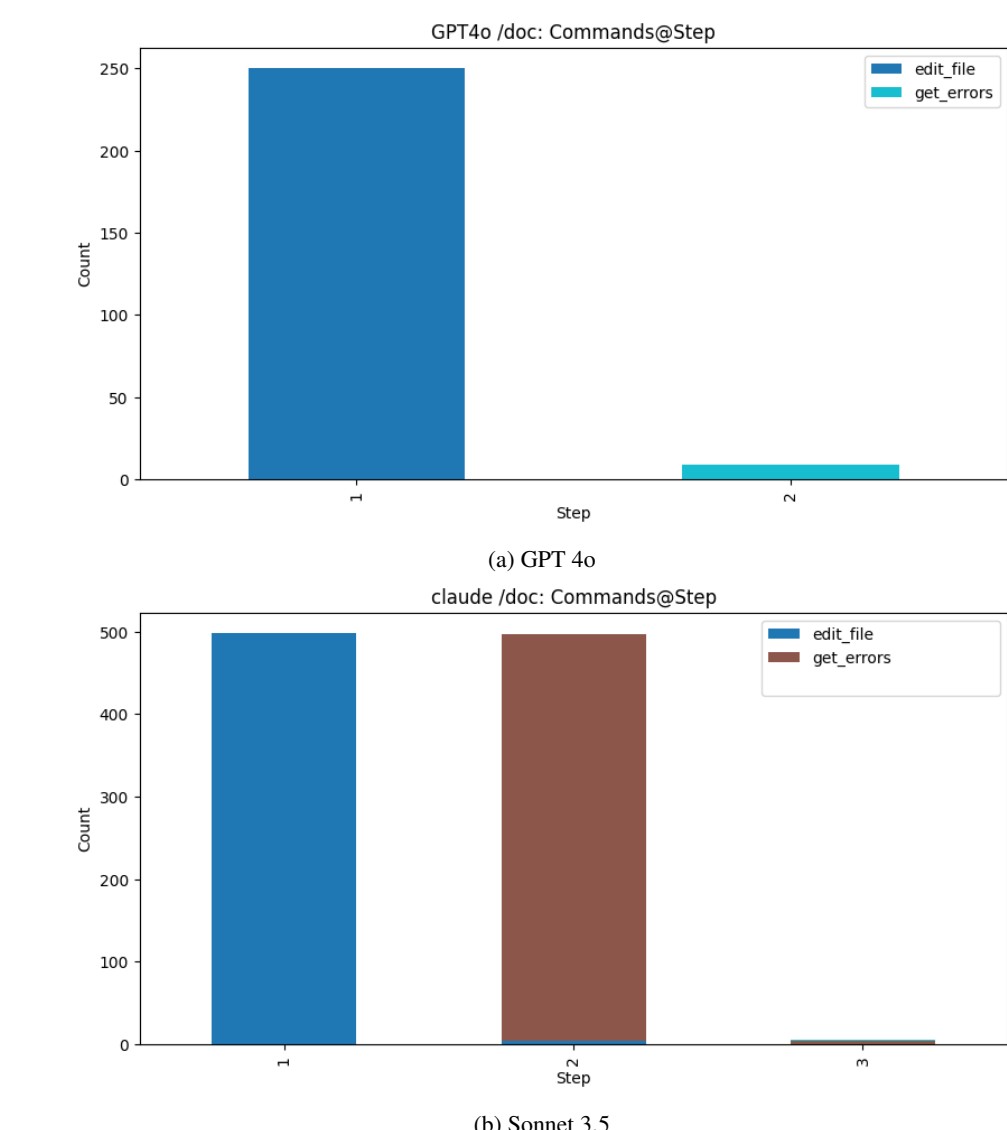

(a) GPT 4o

(b) Sonnet 3.5

Figure 6: GPT4o vs. Sonnet 3.5 agentic trajectories for the docstring generation scenario.

**Evaluating LLMs for Software Engineering Tasks.** One of the most comprehensive works is the paper Hou et al. (2023), which provides a systematic literature review on the intersection of LLMs and SE, covering various aspects such as data collection, pre-processing, application, optimization, evaluation, and prompt engineering. The paper also identifies the current challenges and future directions for LLMs for SE. One of the first works for evaluating LLMs for code considering code execution and test cases is HumanEval Chen et al. (2021), a benchmark dataset and a challenge for measuring the functional correctness of LLMs trained on code. HumanEval consists of 164 hand-written programming problems in Python, each with a function signature, a docstring, a body, and several unit tests. Since then, there have been many augmentations to HumanEval, covering language expansion Athiwaratkun et al. (2023), Cassano et al. (2022), novel completion tasks Muennighoff et al. (2024), and more rigorous testing Liu et al. (2023).

SWE-bench Jimenez et al. (2024) is another standard benchmark for code generation. It uses real-world PR data to create tasks that require LLMs to find files in a repository and apply patches to resolve a given git issue.

In our research, we build upon the foundation laid by prior works in the literature. Like HumanEval and SWE-bench, we incorporate considerations of code execution and test cases, but extend in terms of the breadth of SE tasks addressed and the refinement of evaluation metrics. Furthermore, our emphasis is on developing a comprehensive evaluation framework for LLM-guided programming within IDE interactions, with a particular focus on improving user experience and satisfaction with chat-IDE products.

## 5 CONCLUSION AND FUTURE WORK

**Future Work.** Currently, we are in the process of updating the data for the method generation task. Future work will include results for this task across all five languages. We also plan to continuously add new evaluations to our benchmark, such as query resolution and multiple-file edits and bug fixes.

**Conclusion.** With the growing use of LLMs to aid developers in complex engineering tasks comes the need for more robust evaluations of LLM-generated code. Especially as more companies and products seek to integrate LLMs into their workflows, existing evaluation metrics are not sufficient to confirm the quality and correctness of machine-generated code. In this paper, we propose a solution to this problem via the Copilot Evaluation Harness. We define four key evaluation metrics for the code generation problem space: method generation, test generation, docstring generation, and bug fixing. We detail the methodology required to collect test cases and evaluation results for each of those four metrics. We also provide results for three of the four metrics across myriad programming languages.

Our goal in creating the evaluation harness is to validate the quality of LLM-generated code. Since our benchmark relies on test cases created from hundreds of real-world repositories, it reflects the reality of customer code, and allows engineers to optimize their IDE-LLM integration with that experience in mind. Although we have seen immense advancements in the code generation ML space, we seek to highlight how much oversight and engineering effort is required to reliably integrate LLMs into a code workflow. We aim to provide developers a comprehensive evaluation suite, with which they can optimize LLM integrations. With the Copilot Evaluation Harness, programmers can systematically and robustly evaluate the impact of parameters such as prompt wordings, changes in the order of information provided and changes in the context provided to the model. This is enabled by the interpretability and explainability of results from our metric. In using the Copilot Evaluation Harness, engineers can increase user trust in the quality of code generations.

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

## A APPENDIX

## B DATA VALIDITY

Although our dataset are pulled from real-world git repositories, this does not guarantee that our test cases accurately reflect how users interact with LLMs through IDEs. To confirm the validity of our dataset, we gather usage data that illustrates how hundreds of developers at a major software company use the docstring generation and bug fixing functionalities of our target LLM powered chat extension in VS Code. We then compare these instances to our test cases.

For test generation, we use OpenAI's ada embedding model ada to embed the tests generated from user requests, and compare those to snippets generated from our evaluation cases. Similarly, for the bug fixing telemetry, we embed the code snippets that contain the bug. We use PCA dimensionality reduction to plot the data in two dimensions. PCA dimensionality reduction is optimized to find a plane that maximizes the distance between points and outliers. Figures 7 and 8 show the results of this comparison. We see that each language forms a cluster, and the real usage and our data exist within a similar space for each language cluster, for both test generation and bug fixing.

We do not aim to match the test cases in our dataset to the real usage point for point. Rather, we are determining whether or not our test cases are outliers in the space of the real usage. If they are not outliers, we can infer that our dataset is in line with the real-world usage of the chat extension. This analysis suggests that our dataset for both the test generation and bug fixing evaluation is in line with real world usage.

## C LANGUAGE-SPECIFIC BENCHMARK CREATION DETAILS

For each language, we sample from Github public repositories whose code we are able to build and whose test suites we are able to run using our build agent. The build agent supports Node 18+, Python 3.8+, Java JDK 1.8 (requiring Maven), and .NET 6.0, 7.0 and 8.0.

### C.0.1 JAVASCRIPT AND TYPESCRIPT

In Javascript and Typescript, we sub-select on repos that contain a $package.json$ file at the root directory. The $package.json$ file works in concordance with npm (Node Package Manager) to handle various tasks within the repo, such as specifying dependencies for installation and running the test suite. We rely on npm for our evaluation of Javascript and Typescript code, so we only consider repos whose infrastructure is built to be managed with npm.

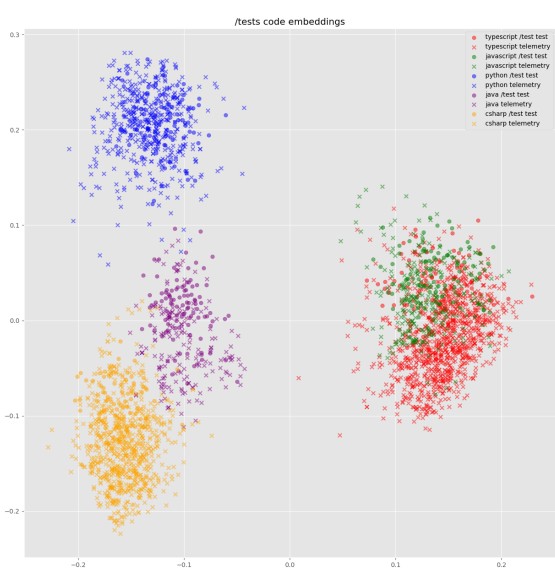

Figure 7: Comparing our dataset for Test Generation evaluation with real-world usage across languages. This diagram shows that the data is clustered by language, not our test cases vs. user cases. This shows that model responses from our test generation test cases align with the model responses from user queries.

### C.0.2 JAVA

In Java, we consider repositories that leverage Maven for their build process. In addition, as of writing, we only consider projects that use JDK 1.8.

### C.0.3 PYTHON

In Python, we only consider repositories for which we are able to successfully install all dependencies within a virtual environment.

## D  METRIC EVALUATION DETAILS

### D.1  DOCUMENTATION GENERATION FROM CODE (DOC)

This task involves generating documentation for a method.

### D.1.1  METRICS

In this scenario, we consider a docstring generation to be successful if the location, format, and coverage of the generated text is correct. We report the following metrics for this scenario:

- *Syntax Correctness*: We check that the docstring has been inserted into the code in such a way that it does not disrupt the syntax of the file with its addition.

- *Format Correctness*: If the documentation comment is placed in a syntactically acceptable manner for the given language, we further check for the correctness of documenting the return statement, function arguments with their types, function name, and whether a function description was written.

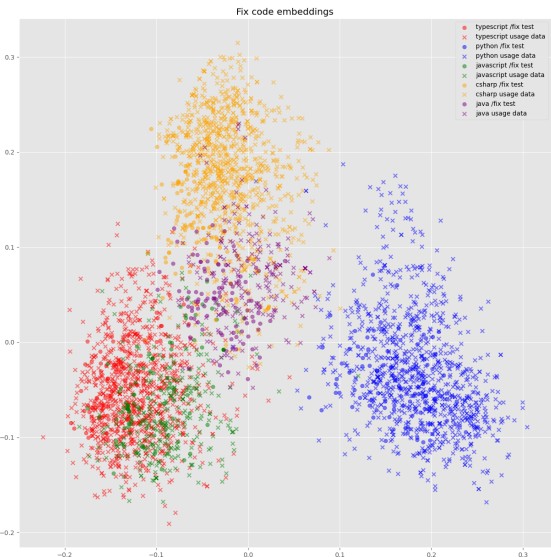

Figure 8: Comparing our dataset for Bug Fixing evaluation with real-world usage across languages. This diagram shows that the data is clustered by language, not our test cases vs. user cases. This diagram shows that the model responses from our bug fixing test cases align with the model responses from user queries.

### D.1.2 EVALUATION PROCEDURE

We begin with a set of methods. For each method, we provide the method's signature and body to the LLM as context. We then prompt the LLM with a request to generate documentation for the method, and return the input function with the generated docstring inserted in the correct location within the function.

After the LLM generates the documentation and the generated docstring is inserted into the code file, we evaluate the syntax correctness of the file with the generated docstring, as well as the correctness of the docstring itself.

### D.2 BUG-FIXING (FIX)

This task involves using LLMs to fix bugs identified by static analysis tools, with an expectation that the resulting fixed code will have fewer errors overall than the original code. We use the following static analyzers:

- javascript: eslint esl;
- ts: eslint esl, tsc (typescript compiler);
- python: pylint pyl, pyright pyr;
- java: spotbugs spo;
- c#: roslyn ros;

If the original error is fixed but another error is introduced in its place, the test case will fail.

### D.2.1 METRICS

In this scenario, we consider a bug fix to be successful if the resulting code is syntactically correct and the corresponding static analysis warning or error has disappeared.

- *Syntax Correctness*: we confirm that the code file with the bug fix remains syntactically correct.
- *Fix Rate*: we check that an existing static analysis warning or error in the code has been successfully resolved by the suggested changes, without introducing any other errors.

### D.2.2 EVALUATION PROCEDURE

Given a set of bugs found by static analyzer tools, we provide the file contents and diagnostic information to the LLM to generate a fix. We assess whether the model fixed the original error, whether it created any new errors, and whether the model-modified code remained syntactically correct after the fix was inserted.

## D.3 CODE GENERATION FROM NATURAL LANGUAGE (GENERATE)

This task involves generating a code snippet from a natural language description.

### D.3.1 METRICS

Similar to previous evaluations of code generations Chen et al. (2021), we consider a generated code snippet to be successful if the generated code is syntactically correct and all test cases covering the generated code pass. Therefore, we report the following metrics for this scenario:

- *Syntax Correctness*: We compute and report the percentage of generated code that is syntactically correct. For this metric, we check the syntax correctness of the generated code using a language-specific parser (e.g., tree-sitter for each language).
- *Test Pass Rate*: We check the number of passing and failing tests and compute the passing test ratio. To compute this number, we execute the entire test suite of the user project and track which tests fail that passed prior to the model's code injection.

### D.3.2 EVALUATION PROCEDURE

We begin with a set of repositories with test cases. From each repository, we select the methods that are: 1) covered by the test cases in the given repository's test suite, and 2) have a docstring. For each method, we ask an LLM to generate the body of the method given the method's signature and docstring. We provide the contents of method's file as context to the LLM, replacing the original method body with a commented line reading "Your Code Here."

After the LLM generates the method body, we put the generated code back in place of the original method body and evaluate the code by running the repository's test suite against the new method body. We then compute and report the syntax correctness and test pass rate, as explained above.

## D.4 TEST CASE GENERATION FOR CODE (TEST)

This task involves using LLMs to generate test cases for code. Developers usually shortcut when it comes to writing unit tests. Automating test generation can motivate more developers to include unit tests.

### D.4.1 METRICS

In this scenario, we consider a generated test to be successful if it can pass on execution. Note that, for this evaluation, this means we assume the code for which the test was written is correct.

- *Generated Test Pass Rate*: We compute the pass rate of the generated test. We assume the original method is correct, and execute the generated test on its focal method.

### D.4.2 EVALUATION PROCEDURE

Given a set of methods, we provide the method signature, docstring, and body as context to the LLM to generate a test for each focal method.

Once the LLM generates a test for the method, we add the test to the repository containing the method, and attempt to execute the test.

For Javascript and Typescript, we generate tests using either the Jest or Mocha library. The original test suite of the repository does not need to be written with either library, but each method's original file must be able to pass without errors when a trivial test case (which essentially just asserts true) is appended to the file. When evaluating the generated tests, we temporarily append them to the focal method's file to mitigate import errors, and run the entire file. If running the file with a trivial test case appended (e.g. a test that should always be true) returns false or an error, we know the results from the generated test on that file are not reliable.

For Python, we write the generated test to a file located in the same directory as the file containing the focal function. We pre-process to only test methods we can successfully import into the test file, so that we know tests are not failing because of import errors over which the model does not have control.

## E    MODEL COMPARISON SUPPLEMENTARY FIGURES

```
This diagnostic has some related code:
```
function( obj ) {
    var name;
    for ( name in obj ) {
        return false;
    }
    return true;
}
```
```

```
isEmptyObject: function( obj ) {
    var name;
    for ( name in obj ) {
        return false;
    }
    return true;
},
```

(a) Code included at the end of `fix` prompt.     (b) True code from the file.

Figure 9: Since the diagnostic does not include the name of the function, it does not align correctly when the model attempts to apply its patch.

