# OpenReview forum: "Copilot Evaluation Harness: Building User Trust in LLMs and LM Agents for IDE Environments"
_ICLR.cc/2025/Workshop/BuildingTrust — Submitted to BuildingTrust_

### Official Review · Reviewer_EVPA · 2025-03-01
**Review to Copilot Evaluation Harness**

**Rating:** 4
**Confidence:** 5

**Review:**

This paper presents an evaluation harness for LLMs on tasks relevant to interactive software development aided by LLMs: generation of documentation, bug fixing, function generation and test case generation. The authors compile a dataset based on 300 repositories across several programming languages and demonstrate that models have strongly varying performance across these tasks.

The topic of the paper is interesting. The gap between HumanEval and similar code generation benchmarks and SWE-bench and similar repository based generation tasks is huge, with the proposed benchmark providing a promising middle-ground.

While the overall setting is promising, I think the work presented is not ready for in-depth discussion and presentation at a workshop. The paper includes several major flaws, including lack of detail about the dataset and used methods and misleading wording. Presentation of the paper also requires much additional work.

First, the paper seems incomplete and lacks crucial details about the evaluation. While the workshop calls for work in progress and negative results, the paper should at least present a detailed and complete explanation of intermediate results.
- The evaluation seems not finished. The authors mention that their setting `generation` is still running and will be added once complete (l.155). I appreciate the honesty but recommend to omit introducing any settings  where no results can be presented (as these can not be peer-reviewed!)
- The current descriptions leaves me with many open questions.
  - The authors mention they evaluate "within [...] Visual Studio Code" (l.73). It is not clear to me how this is implemented nor what benefit this brings (as opposed to evaluating in any sufficient developer-setup like setting).
  - I have trouble understanding how exactly the evaluation integrates "user interaction", Figure 1 is not really helpful in explaining how it works.
  - There are no details on the exact composition of the final dataset (introduced in sec 2.1). How many repositories remain after filtering? How many functions are picked for doc/test/generate/fix, how long are they on average, are they highly integrated into the environment (many external calls) or isolated (like HumanEval)? Which kind of static errors (statistics about this) are in the fix setting (how many are resolved by LLMs/which ones)?
  - How is the "coverage" of generated documentation measured? This does not seem like a standard term to me (for documentation)
  - Do you only consider strict decrease of errors in the fix setting or *also* make sure that the original error disappears? I can imagine that the LLM fixes some formatting related errors but does not actually resolve relevant errors?
  - Do you maintain only the function signature for the generate setting or do you also check that documentation is present?
  - How are model patches applied? Do models have to generate patches in a one-shot setting?
- The described coding agent in line 115 seems very interesting and I recommend the authors to provide more detail on how and how well it works.

Moreover I think there is actually no interaction in any of the setups (chat or agentic) except for the initial prompt, so how is the harness different to HumanEval/SWE-Bench in this respect (compare l. 69)? If there is just a single prompt, please refer to it this way and don't mention "user interaction" (e.g. Figure 1). The benchmark seems to evaluate LLM performance on Copilot-setting user prompts but not on user interaction, which implies several rounds of editing with user feedback, as done by e.g. Cursor or Aider. LLM-guided coding (as mentioned in the abstract) also seems to be imprecise, as the LLM is still guided by a user prompt and not a user guided by the LLM.



Nitpicks:
- For the setting of test generation, SWT-bench and LIBRO seem like related work [1,2]
- The test setting appears to be very difficult for code models and agents. This appears surprising as test generation in SWT-Bench [1] appeared to be similarly difficult as patching in SWE-bench.
- Please provide averages for languages and LLMs in Table 1/2. Other works [3,4] usually show results with rows per LLM, and I recommend sticking to this formatting to avoid confusion among readers.
- Please correctly use citep in line 36 and similar settings
- You mention that the eval harness "enables a new level of understanding of model behavior [beyond numeric values of other benchmarks]" (l.68). Why? The main results of your benchmark are numeric values (Table 1/2).
- Figure 2-5 provide too much detail for the main text. I recommend providing only a summary of the results and moving the figures to the appendix.
- l153 you refer to 4 LLMs as "myriad". This term is already very informal and I don't think it fits either (I would definitely expect > 4 evaluated LLMs for this word)

[1] Mündler et al. *SWT-Bench: Testing and Validating Real-World Bug-Fixes with Code Agents*
[2] Kang et al. *Large Language Models are Few-shot Testers: Exploring LLM-based General Bug Reproduction*
[3] Chen et al *Evaluating Large Language Models Trained on Code*
[4] Jiminez et al. *SWE-bench: Can Language Models Resolve Real-World GitHub Issues?*

---

### Official Review · Reviewer_bjs7 · 2025-03-01

**Rating:** 10
**Confidence:** 4

**Review:**

This work proposes a new benchmark / harness along the lines of SWE-Bench, but testing a more general capability: interactive IDE scenarios.

I find it significant because AFAICT there are no other benchmarks that test this realistic user scenario. SWE-Bench has a much more restricted setting. General purpose coding assistants will need to be integrated with a user's development environment, rather than being restricted to make PRs based on well-defined bug requests / issues.

Each of the scenarios evaluated makes sense, and I think the bug-fixing scenario is particularly interesting given the access to static analyzer errors. I find that existing models are poor at dealing with static analyzer errors. Finally, the analysis of existing models and their sensitivity to instructions is an interesting and useful empirical finding.

---

### Official Review · Reviewer_sH7b · 2025-03-02
**Great step forward for LLM+Code evals but lacking rigor**

**Rating:** 5
**Confidence:** 5

**Review:**

This paper proposes a new evaluation harness with new metrics that increases breadth as compared to previous works. It proposes 4 different tasks in two different settings (single turn vs multi-turn): 1) document generation, 2) bug fixing, 3) code generation, and 4) test case generation. As it correctly points out, existing popular benchmarks like HumanEval and SWE-Bench are more limited in what they are measuring. However, while the paper does increase breadth, it lacks some depth and fails to acknowledge it at several key points. Furthermore, some important design decisions and technical details are missing. Nonetheless, the initial experiments show some interesting insights which will undoubtedly play a role in 2025 if they do indeed hold true. That is, reasoning models are a poor fit in agentic settings.

- The paper mentions that 5 and 6 languages are used. It seems only 5 were evaluated so that should be updated.
- Repositories are filtered by size & in "generate" settings, "correctness is measured by running the repository's test suite". There is a critical oversight in that the ratio of tests to lines of code or separately, the code coverage of repository is not considered.
- A build agent is used throughout the evaluations and details of this agent are left out. It's unclear if any evaluation errors could have been from this agent.
- In the "doc" setting, there are essentially syntactic checks but no semantic checks. This oversight is not even acknowledged.
- In "fix", the authors mention a tradeoff between selecting from two different metrics but fail to consider that both could be possible at the same time.
- It would be helpful if the best model per row in the tables were bold
- The chat problem formulation is contradicting. In the problem formulation, it says only a one line query is specified but in the next paragraph, we see that the bug scenario uses few-shot prompting. This raises questions about the overall insights.
- In the agentic scenario, the terminal commands are not specified.
- In related works, "Unlike traditional machine learning models where k-fold cross validation was a common evaluation process, LLMs are often evaluated using static data sets". I dont follow this statement. Computer vision uses other metrics and even within NLP, perplexity is still used. There are also some new ones like false refusal rates or verbatim usage across n-grams.
- SE is not defined

Given the shacky evaluations and oversights, this paper has potential but is not read as is.

---

### Decision · Program_Chairs · 2025-03-05

**Decision:**

Reject

**Comment:**

As noted by the three reviewers, the topic is interesting. However, the execution is incomplete, with unclear dataset composition, missing technical details, and contradictions in problem formulation. Additionally, the evaluation is still in progress, raising concerns about the readiness of the work for publication. While one reviewer rates the paper highly, the majority highlight significant gaps. Given the paper's lack of completeness, I recommend rejection, with the suggestion to refine and resubmit when the evaluation is more robust.